environmental chemistry

hydrochar, cornstalk, mesotrione removal, adsorption

**Authors for correspondence:**
Jingmin Yang
e-mail: yangjingmin@jlau.edu.cn
Haibo Chang
e-mail: changhb@jlau.edu.cn

[†]This author contributed equally to this work and is co-first author of the paper.

This article has been edited by the Royal Society of Chemistry, including the commissioning, peer review process and editorial aspects up to the point of acceptance.

# Preparation and characterization of cornstalk microspheric hydrochar and adsorption mechanism of mesotrione

Zhongqing Zhang, Mengmeng Zhou[†], Jinhua Liu, Jiahao Li, Jingmin Yang and Haibo Chang

College of Resources and Environmental Science, Jilin Agricultural University, Changchun 130118, Jilin, People's Republic of China

JY, 0000-0003-2551-2259

In this study, cornstalk was pyrolysed to obtain hydrochar (HC), which was used to remove mesotrione from aqueous solutions. HC characterization and batch experiments were conducted to investigate mesotrione adsorption and the underlying mechanism. The characterization revealed microspheres on the HC surface. FT-IR spectra showed that the HC contained a large number of –OH groups, C=C bonds of aromatic rings, C–H groups in aromatic rings and phenolic C–O bonds. The adsorption results showed that the mesotrione adsorption ability gradually increased as the HC preparation temperature increased. The quasi-second-order kinetic equation ($R^2 \geq 0.9860$, $p < 0.05$) agreed well with the mesotrione adsorption process. The maximum monolayer adsorption capacity, which was obtained at pH 7 and 45°C with HC prepared at 240°C, was 3181.7 mg kg$^{-1}$ with the Langmuir isotherm model ($R^2 \geq 0.9491$, $p < 0.05$). Van der Waals and dipole forces and hydrogen bonds were inferred as the main adsorption mechanisms. HC has potential as an effective and energy-saving adsorbent for mesotrione to reduce environmental pollution.

## 1. Introduction

In recent years, with the increase in intensive agricultural production, many herbicides have been applied in agriculture, leading to a decline in soil and agricultural product quality and intensification of agricultural non-point-source pollution, threatening the safety of the food chain [1]. Mesotrione has been widely applied in agriculture due to its high activity, low

toxicity and ability to be easily mixed. Mesotrione has mainly been applied for annual broad-leaved weeds and some grasses pre- and post-emergence in maize fields [2]. Additionally, mesotrione could significantly impact surface water and groundwater due to its strong mobility in soil [3]. Other study reported mesotrione was stable under natural light and hardly photolysed, the degradation half-life was as long as 84 days in aqueous solution and the photolysis half-life was 15–21 days in soil [4]. Undegraded Mesotrione would have potential influence on water environment pollution when it enters water body.

Carbonaceous materials prepared from agricultural wastes absorb agricultural chemical pollutants, providing a new approach for the utilization of agricultural wastes. Pyrolysis [5], gasification [6] and microwave pyrolysis [7] are the main preparation methods for such products. In contrast with pyrolysis, hydrochar (HC) is produced with water at 180–375°C under autogenous pressure and has been used to treat biowastes due to the low temperature required for its production and associated energy savings. HC is an economically friendly, flexible and highly efficient technology to remove pesticide residues [8], heavy metals [9] and antibiotics [10] from wastewater and soil. Eibisch *et al*. [11] noted that the application of HCs to agricultural soils provides an effective management practice to remove the pesticide isoproturon while also reducing the risk of drinking water contamination. Kambo & Dutta [12] indicated that the physico-chemical properties and application effects of HC were superior to those of pyrolysed biochar.

In China, more than 7 million metric tons of cornstalk are produced every year [13]. Most of them are burned and discarded, which caused a series of environmental problems. Thus, it is necessary to find an economical and effective approach to address this issue to improve environmental quality. The objective of this study was to select a high-capacity HC to adsorb mesotrione and study the effects of the initial pH of the solution, time, ionic concentration and temperature on the mesotrione sorption of HC.

# 2. Experimental section

## 2.1. Preparation of hydrochar

The cornstalks were collected from Agricultural Science Experimental Station of Jilin Agricultural University. The cornstalks were dried (at 20°C), pulverized and passed through a 70 mesh sieve. First, 3 g cornstalk was mixed with 60 ml distilled water at a 1 : 20 solid–liquid ratio. The mixture was stirred for 30 min at room temperature. Then, the samples were sealed in a Teflon-lined stainless steel autoclave at 190–240°C for 24 h under autogenous pressure. Finally, the reactor was cooled to room temperature. The collected HCs were washed with deionized water and dried at 60°C for 12 h. The HC samples prepared at 190°C, 220°C and 240°C were labelled HC-190, HC-220 and HC-240, respectively.

## 2.2. Characterization of materials

The surface morphologies of the HCs were observed using a field emission scanning electron microscope (SEM, Tecnai-G2F30, Ltd, Japan). The surface functional groups of the samples were determined using Fourier transform infrared spectroscopy (FT-IR, Vertex 70, Germany). The concentration of mesotrione was measured by HPLC (Agilent Technologies 1260 Infinity, USA).

## 2.3. Adsorption experiments

### 2.3.1. Batch equilibrium studies

Batch adsorption equilibrium experiments were performed in a 50 ml centrifuge tube containing 12 ml 30 mg l$^{-1}$ mesotrione solution with 0.02 g HC-240 in a temperature-controlled shaking water bath at 298, 308 and 318 K. The optimum initial pH (PHS-3C) of the solution of mesotrione was identified by carrying out 24 h batch adsorption with mesotrione, with the pH adjusted from 3 to 11 using 1 M NaOH and 1 M H$_3$PO$_4$. The optimum initial pH was used in subsequent experiments. Batch adsorption isotherms were obtained using six mesotrione initial concentrations (0, 10, 20, 30, 40, 50 mg l$^{-1}$) at pH 7 for 2 h at 298 K. All batch sorption experiments were performed in at least two replicates. All samples were centrifuged before analysis of the mesotrione concentrations. The average results were calculated with the standard errors. The adsorption data of the HC were analysed by

Origin software. The amount of mesotrione adsorbed by the HC was calculated according to the below equation

$$Q_e = \frac{(C_0 - C_e)V}{m}, \tag{2.1}$$

where $Q_e$ (mg g$^{-1}$) is the amount of mesotrione adsorbed; $C_0$ (mg l$^{-1}$) and $C_e$ (mg l$^{-1}$) are the initial and equilibrium concentrations of mesotrione, respectively; $V$ (l) is the volume of the liquid solution and $m$ (g) is the mass of adsorbent.

### 2.3.2. Batch kinetic studies

The batch kinetic studies of mesotrione at 298 K and pH = 7 were performed with HC-240 following the same process as that for the batch equilibrium study. Samples were collected at different times during the adsorption process to determine the concentration of mesotrione.

The mesotrione was quantified by a Dionex UHPLC 3000 (Pursuit XRS 5, C 18, 4.6 × 150 mm). A mixture of 0.1% phosphoric acid and methanol was applied as the mobile phase. The injection volume was 20 µl with a 45 : 55 volume ratio, and the flow rate was 1 ml min$^{-1}$. The holding time was 6 min with a mesotrione retention time of 4.5 min. The detection wavelength was 270 nm. The experiment was repeated three times under the same conditions, and a blank experiment was performed using the same procedure without mesotrione.

# 3. Results and discussion

## 3.1. Characterization of the prepared hydrochar

The HC samples were characterized by SEM imaging, and the morphologies and structures of HC are shown in figure 1. The surface of the HC maintained a completely smooth and layered morphology with preparation at 190°C (figure 1a), whereas many small bumps and a rough texture became apparent with preparation at 220°C (figure 1b), showing the initial stages of microsphere formation. The HC exhibited dense, broken and numerous microspheres on the loose sheet surface with preparation at 240°C (figure 1c). These SEM images illustrated that the HC was not entirely decomposed during the low-temperature hydrothermal process. The microsphere structure was produced at high temperature. Additionally, the high temperature was beneficial to the hydrothermal reaction in cornstalks [14]. The hemicellulose and cellulose in the cornstalk rapidly underwent polymerization, dehydration and carbonization. The carbonization reaction was completed, and polymerization occurred at the same time. Subsequently, the HC microspheres formed [15].

The hydrothermal carbonization process was seriously influenced by temperature. Therefore, infrared spectroscopy analysis of the HCs that were produced at different temperatures was performed. The FT-IR spectra of HC are presented in figure 2. The characteristic absorption bands appeared at approximately 3340 cm$^{-1}$ and were assigned to inter/intrahydrogen bonded –OH stretching vibration absorption peaks [16]. The intermolecular or intramolecular dehydration of the cornstalks increased with temperature. The –OH stretching vibration was attributed to lignin and cellulose components. The peak at 2935 cm$^{-1}$ was caused by symmetric and asymmetric stretching vibrations of the –CH$_3$ and –CH$_2$ of aliphatic hydrocarbons or cycloalkanes. The –CH$_3$ and –CH$_2$ stretching vibrations were stable in the HC over the short term, and demethylation reactions did not occur [17,18].

Additionally, the absorption peaks at 1760 and 1608 cm$^{-1}$ were characteristic of the C=O stretching vibration absorption of hemicellulose acetyl groups [19]. The absorption peak gradually decreased with high temperature. The decarbonylation reaction produced O$_2$. Both spectra had peaks at 1518 and 1450 cm$^{-1}$ corresponding to the C=C stretching vibration of the aromatic ring skeleton structure in the HC [20]. The aromatic ring skeleton structure mainly existed on lignin, and the lignin still retained the complete aromatic structure under the low hydrothermal temperature. Another band between 1250 and 1000 cm$^{-1}$ was assigned to the ring vibrations and the stretching of C–OH side groups and the C–O–C and C–O glycosidic bond vibrations of celluloses and hemicelluloses [21]. The functional groups on the surface of the HC played an essential role in adsorbing mesotrione.

## 3.2. Effect of pH on the adsorption of mesotrione onto hydrochar

To determine the effect of the initial pH of the solution on the adsorption of mesotrione onto HC, batch experiments were undertaken at 298 K by adjusting the initial pH of the solution from 3 to 11 while

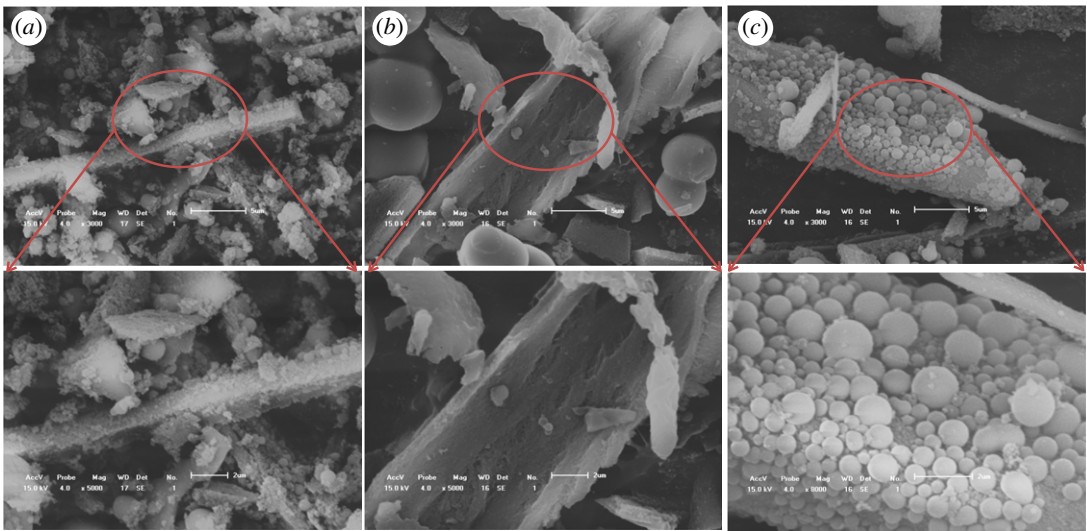

**Figure 1.** Scanning electron microscopy (SEM) images of HC; (*a*) HC-190, (*b*) HC-220 and (*c*) HC-240.

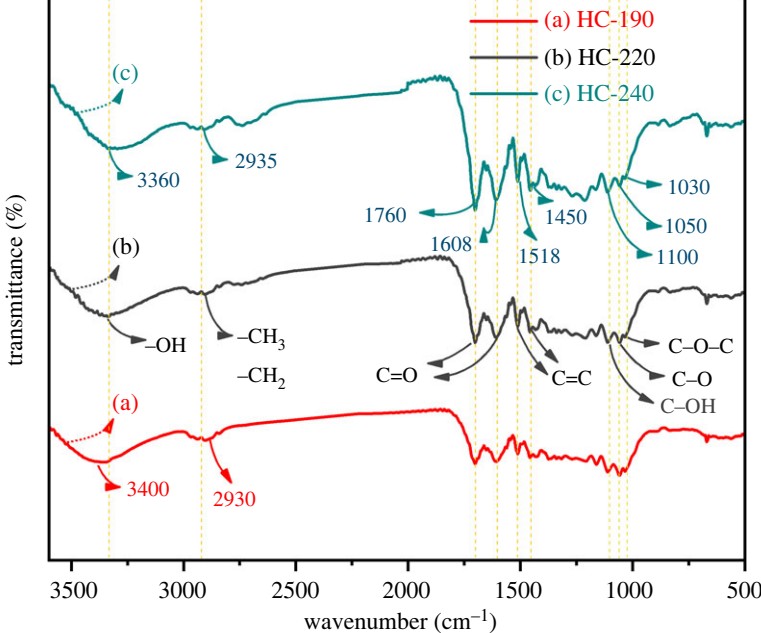

**Figure 2.** FT-IR characterization of HC; (a) HC-190, (b) HC-220 and (c) HC-240.

keeping the other conditions constant; the results for HC-240 are plotted in figure 3. The maximum and minimum adsorption capacities were 3.0138 mg g$^{-1}$ (pH=3) and 0.5093 mg g$^{-1}$ (pH=9), respectively. The adsorption capacity and removal rate rapidly reduced when the initial solution pH rose from 3 to 9.

## 3.3. Effect of temperature on adsorption

Temperature was an important factor affecting the sorption process. The influence of temperature on the adsorption capacity of the HC was studied at pH=7. The sorption behaviours were investigated by plotting $Q_e$ against $C_e$ (figure 4). The adsorption of mesotrione at low concentrations, $C_0$ (0–20 mg l$^{-1}$), increased quickly from 0 to 1.8, 0 to 2.5 and 0 to 3.2 mg g$^{-1}$ for HC-190, HC-220 and HC-240, respectively, and then levelled off when the mesotrione concentrations were beyond 30 mg l$^{-1}$ (figure 4). Figure 4 also demonstrates that the adsorption of mesotrione was not affected by temperature. Divincenzo & Sparks [22] researched the adsorption of pentachlorophenol ions and molecular formation in soils at different temperatures (277, 288 and 328 K). They found that the adsorption of

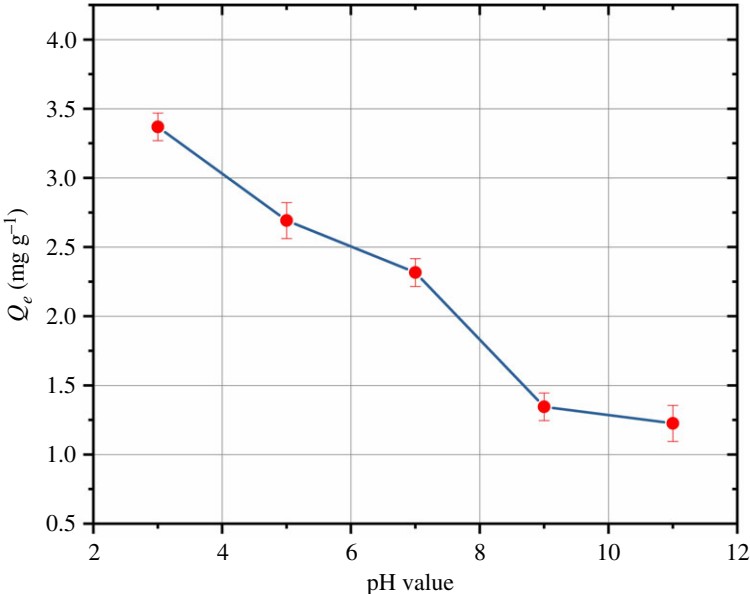

**Figure 3.** Adsorption of mesotrione by HC-240 at different initial pH values.

**Figure 4.** Adsorption of mesotrione by HC at different preparation temperatures.

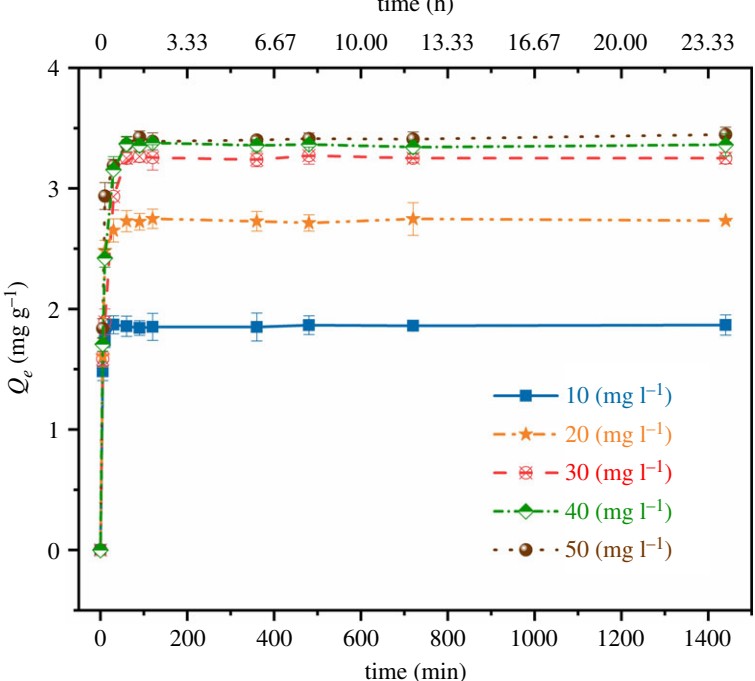

**Figure 5.** Adsorption of mesotrione by HC at different preparation temperatures.

pentachlorophenol molecules did not change with temperature. Similar results on the adsorption of 2.4-dichlorophenoxy propionic acid herbicide in soil were found by Thorstensen *et al*. [23]. Yang [24] found that the adsorption of mesotrione by biochar fired from pomelo peel was not affected by the reaction temperature, and there was no obvious rule with the increase in reaction temperature.

## 3.4. Adsorption kinetics

Quasi-first-order and quasi-second-order kinetic models were used to estimate the rate of the adsorption process. The contact time required for the adsorption of mesotrione by HC to reach liquid–solid equilibrium is presented in figure 5. The maximum adsorption capacities of the HC at equilibrium were 1.947, 3.015, 3.442, 3.564 and 3.694 mg g$^{-1}$, corresponding to initial mesotrione concentrations of 10, 20, 30, 40 and 50 mg l$^{-1}$, respectively.

The adsorption process of HC on mesotrione was divided into three stages: the fast adsorption stage, slow adsorption stage and adsorption equilibrium stage. This phenomenon was due to the abundant microsphere structure on the surface of the HC, providing a large number of adsorption sites for mesotrione in the initial stage, which resulted in a significant increase in the adsorption capacity [23]. After 30 min, the adsorption of mesotrione steadily slowed down, which might have been due to the saturation of the adsorption sites on the surface of the HC and the resistance of mesotrione to diffusion into the interior of the HC.

Kinetic data were fitted to three kinetic models: quasi-first-order and quasi-second-order models, as shown in equations (3.1) and (3.2).

Quasi-first-order model

$$q_t = q_e[1 - \exp(-k_1 t)] \tag{3.1}$$

and quasi-second-order model

$$q_t = \frac{k_2 q_e^2 t}{1 + k_2 q_e t}, \tag{3.2}$$

where $t$ (min) is the adsorption time, $q_t$ (mg g$^{-1}$) is the amount of mesotrione adsorbed onto the HC at time $t$, $q_e$ (mg g$^{-1}$) is the amount of mesotrione adsorbed at equilibrium, $k_1$ and $k_2$ are the rate constants of the quasi-first-order and quasi-second-order adsorption models, respectively.

**Table 1.** Kinetic parameters of mesotrione adsorption onto HC-240.

| initial concentration (mg l$^{-1}$) | $q_{e,exp}$ (mg g$^{-1}$) | quasi-first-order | | | | | quasi-second-order | | | |
|---|---|---|---|---|---|---|---|---|---|---|
| | | $q_{e,cal}$ (mg g$^{-1}$) | $k_1$ (min$^{-1}$) | $R^2$ | RMSE (%) | | $q_{e,cal}$ (mg g$^{-1}$) | $k_2$ (g mg$^{-1}$ min$^{-1}$) | $R^2$ | RMSE % |
| 10 | 1.947 | 1.852 | 0.314 | 0.996 | 2.182 | | 1.896 | 0.465 | 0.999 | 1.171 |
| 20 | 3.015 | 2.839 | 0.197 | 0.974 | 2.604 | | 2.929 | 0.139 | 0.980 | 1.276 |
| 30 | 3.442 | 3.339 | 0.108 | 0.971 | 2.819 | | 3.343 | 0.054 | 0.986 | 1.286 |
| 40 | 3.564 | 3.379 | 0.133 | 0.984 | 2.823 | | 3.390 | 0.066 | 0.999 | 2.180 |
| 50 | 3.694 | 3.427 | 0.133 | 0.971 | 3.232 | | 3.513 | 0.097 | 0.994 | 2.191 |

The kinetic models were fitted based on the experimental data. The model parameters and the calculated and measured $R^2$ and root mean square error (RMSE%) values for the adsorption of mesotrione onto HC are listed in table 1.

Among the kinetic models used, the quasi-second-order model was the best with the largest $R^2$ (0.999), and the equilibrium adsorption value calculated by the model was extremely close to the experimental values. Therefore, the quasi-second-order model best described the adsorption process of mesotrione onto HC.

The quasi-first-order model and quasi-second-order model were evaluated by the relative/ normalized RMSE% [25]. The RMSE% was used to evaluate the models' agreement with the experimental values, as shown in the below equation

$$\text{RMSE\%} = \sqrt{\frac{\sum (q_{e.\exp} - q_{e.\text{cal}})^2}{N}} / q_{e.\exp} \cdot 100\%, \tag{3.3}$$

where $q_{e,\text{cal}}$ and $q_{e,\exp}$ are the calculated and the experimental values, respectively, and $N$ is the number of experiments.

The lower the RMSE% value is, the more suitable the model. The closer the RMSE% value is to 0, the better the model selection and fitting. The RMSE% of the quasi-second-order model was 1.171–2.191, which was lower than that of the quasi-first-order model (2.182–3.232). This demonstrated that the quasi-second-order model was well suited for describing the kinetic characteristics of the adsorption process.

## 3.5. Adsorption isotherms

The application of adsorption isotherms helps to elucidate the interaction between the adsorbent and the adsorbate in the adsorption process. The parameters obtained from different models provided some significant information on the sorption mechanisms. According to previous experimental conclusions, the temperature has no extraordinary effect on the HC adsorption of mesotrione. The temperature of 298 K was chosen in this experiment.

Langmuir, Freundlich and Temkin isotherm plots and linear regression coefficients for the adsorption of mesotrione onto HC-240 at 298 K and pH=7 are presented in figure 6a–c. Adsorption isotherm experimental data were fitted with the three adsorption isotherm models, as shown in equations (3.4)–(3.6), respectively.

Langmuir

$$q_e = \frac{q_m K_L C_e}{1 + K_L C_e}. \tag{3.4}$$

Freundlich

$$q_e = K_F c_e^{1/n}. \tag{3.5}$$

Temkin

$$q_e = \frac{RT}{\beta_T} \ln(A_T C_e), \tag{3.6}$$

where $q_e$ (mg g$^{-1}$) is the amount adsorbed per mass of adsorbent; $q_m$ (mg g$^{-1}$) is the maximum monolayer adsorption capacity; $K_L$ (l mg$^{-1}$) is the Langmuir isotherm constant; $C_e$ is the equilibrium concentration; $n$ is

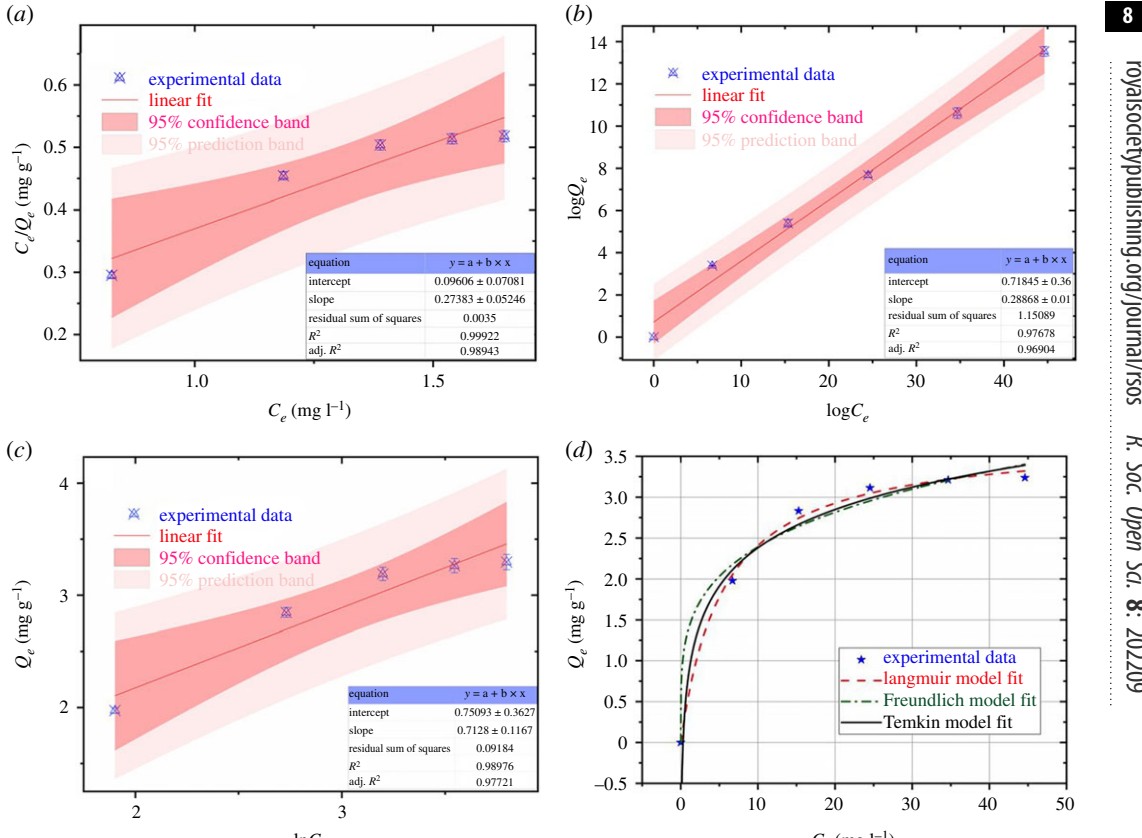

**Figure 6.** Linear isotherm plots and regression correlation coefficients for the adsorption of mesotrione onto HC-240; (*a*) Langmuir, (*b*) Freundlich, (*c*) Temkin and (*d*) adsorption isotherm models of mesotrione onto HC-240 at pH = 7 and temperature = 25°C.

**Table 2.** Adsorption isotherm parameters of mesotrione onto HC.

| adsorbent | Langmuir | | | Freundlich | | | Temkin | | |
|---|---|---|---|---|---|---|---|---|---|
| | $Q_m$ (mg g$^{-1}$) | $K_L$ (l g$^{-1}$) | $R^2$ | $n$ | $K_f$ ((mg g$^{-1}$) (mg$^{-1}$)$^{1/n}$) | $R^2$ | $A_T$ (l g$^{-1}$) | $\beta_T$ (J mol$^{-1}$) | $R^2$ |
| HC-190 | 2.2733 | 0.1202 | 0.9491 | 3.5703 | 0.6693 | 0.9152 | 0.1302 | 0.4788 | 0.9313 |
| HC-220 | 2.9300 | 0.1602 | 0.9856 | 4.1357 | 1.0403 | 0.9627 | 0.5315 | 0.5464 | 0.9718 |
| HC-240 | 3.1817 | 0.1807 | 0.9894 | 4.2630 | 1.4119 | 0.9690 | 0.8235 | 0.6875 | 0.9772 |

a measure of adsorption linearity; $K_F$ is the Freundlich isotherm constant; $\beta_T$ (J mol$^{-1}$) is the Temkin constant related to the heat of sorption; $A_T$ (l g$^{-1}$) is the Temkin isotherm constant; $R$ (8.314 J K$^{-1}$ mol$^{-1}$) is the gas constant; and $T$ (K) is the absolute temperature.

The experimental data plotted with three theoretical model curves (Langmuir, Freundlich and Temkin isotherms) onto HC-240 at 298 K and pH = 7 are presented in figure 6*d*. This also indicated that the Langmuir model best described the adsorption process on mesotrione because the values with this model were close to the experimental values, whereas the values of the Freundlich and Temkin models slightly deviated from the experimental data points.

All of the fitting parameters for HC-190, HC-220 and HC-240 are shown in table 2. The nonlinear regression $R^2$ of the Langmuir model ranged from 0.9491 to 0.9894, compared with the $R^2$ of the Freundlich and Temkin models, which ranged from 0.9152 to 0.9690 and 0.9313 to 0.9772, respectively. Therefore, the three models were suitable for characterizing the equilibrium adsorption process, but the Langmuir model showed the best fit based on $R^2$ and was the most consistent with the experimental data. The results indicated that adsorption by the HC was well characterized by the

**Table 3.** Thermodynamic parameters for the adsorption of mesotrione onto HC-240.

| adsorbent | temperature (K) | $\Delta G$ (kJ mol$^{-1}$) | $\Delta H$ (kJ mol$^{-1}$) | $\Delta S$ (J mol$^{-1}$ K$^{-1}$) |
|---|---|---|---|---|
| HC-240 | 298 | −28.98 | 4.26 | 115.50 |
| HC-240 | 308 | −30.23 | 4.26 | 115.50 |
| HC-240 | 318 | −31.29 | 4.26 | 115.50 |

single molecule absorption model, which was consistent with Dyson *et al.* [26]. In table 2, $K_f = 0.6693$ was calculated by the Freundlich equation for HC-190, while the strongest adsorption capacity of mesotrione was found with HC-240 ($K_f = 1.4119$). The adsorption capacity of the HC steadily increased with the HC preparation temperature (HC-240 > HC-220 > HC-190). The adsorption strength (*n*) of mesotrione on the three types of HC was between 3.5703 and 4.2630. As the carbonization temperature increased, the adsorption strength increased by degrees, and the adsorption nonlinearity gradually decreased. According to the shape of the relationship between the *n* value and the adsorption isotherm [27], the adsorption strength of mesotrione for HC-190, HC-220 and HC-240 was $n > 1$, which corresponds to the 'L-type' adsorption isotherm. This result indicated that the adsorbed solute molecules changed the surface structure of the adsorbent, thereby promoting further adsorption of the solute [28].

In addition, the Temkin model also well described the adsorption process of mesotrione on HC. The correlation coefficient, $R^2$, was between 0.9313 and 0.9772. The Temkin model showed that the chemical adsorption process was based on electrostatic adsorption, which involves interaction between molecules when the HC adsorbed mesotrione [29].

## 3.6. Thermodynamics study

Based on the Langmuir model parameters, the standard Gibbs free energy ($\Delta G$), the standard enthalpy change ($\Delta H$) and the standard entropy change ($\Delta S$) of adsorption were calculated by the below equation (3.7)

$$\Delta G = -RT\ln K \tag{3.7}$$

where $\Delta G$ (kJ mol$^{-1}$) is the change in Gibbs free energy, $R$ (8.314 J K$^{-1}$ mol$^{-1}$) is the gas constant, $T$ (K) is the absolute temperature and $K$ is the distribution coefficient.

The enthalpy change ($\Delta H$) (kJ mol$^{-1}$) and entropy change ($\Delta S$) (J mol$^{-1}$ K$^{-1}$) for the adsorption process were calculated by the below equation

$$\ln K = \frac{\Delta S}{R} - \frac{\Delta H}{RT}, \tag{3.8}$$

where $R$, $T$ and $K$ are the gas constant (8.314 J K$^{-1}$ mol$^{-1}$), the absolute temperature (K) and the distribution coefficient, respectively. $\Delta H$ is the change in enthalpy, and the change in entropy is $\Delta S$. The values of $\Delta H$ and $\Delta S$ were determined by plotting $\ln K$ with an intercept of $1/T$.

Table 3 shows that the negative value of $\Delta G$ for all temperatures of 25°C (298 K), 35°C (308 K) and 45°C (318 K) demonstrated that mesotrione sorption onto HC was a spontaneous reaction [30]. The positive value of $\Delta H$ indicated an endothermic process between the adsorbate and the adsorbent [31]. Mesotrione molecules entered the surface or interlayer of the HC from the aqueous phase. The positive $\Delta S$ indicated that the amount of mesotrione adsorbed on the surface decreased as the degree of freedom decreased. These values are used to study the adsorption of non-polar and polar compounds to soils [32].

## 4. Conclusion

In this study, HC was fabricated from cornstalk by a mild hydrothermal method and used to effectively remove mesotrione from aqueous solution. A higher preparation temperature (240°C) enhanced the capacity of HC to adsorb mesotrione. The microspheres of HC formed from the HC surface and contained a large number of –OH groups, C=C bonds of aromatic rings, C–H groups in aromatic rings and phenolic C–O bonds. The quasi-second-order kinetic equation ($R^2 \geq 0.9860$, $p < 0.05$) agreed well with the mesotrione adsorption process. The maximum monolayer adsorption capacity was

3181.7 mg kg$^{-1}$ based on the Langmuir isotherm model ($R^2 \geq 0.9491$, $p < 0.05$), which was obtained at 240°C, pH 7 and 45°C. The van der Waals force, hydrogen bond and dipole force were inferred as the main adsorption mechanisms. In addition, cornstalk was hydrothermally carbonized, expanding its application value. From an environmental perspective, a more energy-saving preparation method and higher capacity for adsorbing mesotrione are needed. HC should be considered extensively for its broad environmental applications.

Data accessibility. The data are provided as electronic supplementary material.

Authors' contributions. J.Y. and H.C. conceived and designed the experiments. Z.Z. and M.Z. performed the experiments and collected the data. Jin.L. analysed the data. Jia.L. contributed analysis tools. Z.Z. wrote the paper. Z.Z. and M.Z. contributed equally to this work and are co-first authors of the paper.

Competing interests. We declare we have no competing interests.

Funding. This work was supported by the Science and Technology Department of Jilin Province (20190303086SF) and (20200201011JC).

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
