## [Peer Review File · Royal Society Open Science]

Review History

RSOS-202209.R0 (Original submission)

Review form: Reviewer 1

Is the manuscript scientifically sound in its present form?

Yes

Are the interpretations and conclusions justified by the results?

Yes

Is the language acceptable?

Yes

Do you have any ethical concerns with this paper?

No

Have you any concerns about statistical analyses in this paper?

No

Recommendation?

Accept with minor revision (please list in comments)

Comments to the Author(s)

Yang and co-workers reported a mild hydrothermal method to acquire the microspherical structure of hydrochar from cornstalk and investigated the adsorption capacity of hydrochar. The article is well organized and the presentation is also good. However, some minor issues still need to be improved:

1. For the SEM images, the figure 1(c) should provide with the same magnification and scale compared with figure 1(a)(b).
2. The adsorption capacity and removal rate decreased or tended to become stable when $\text{pH} > 9$. Please illustrate why the optimum initial $\text{pH} = 7$ was selected for the subsequent experiments.

Review form: Reviewer 2

Is the manuscript scientifically sound in its present form?

Yes

Are the interpretations and conclusions justified by the results?

Yes

Is the language acceptable?

Yes

Do you have any ethical concerns with this paper?

No

Have you any concerns about statistical analyses in this paper?

Yes

Recommendation?

Major revision is needed (please make suggestions in comments)

Comments to the Author(s)

Grammatical mistakes and spelling:

post-emergence line 46

The claim in line 47 requires reference

The thoughts on line 47 and 48 needs proper connection

The threat of the compound's presence in water has not been highlighted

In line 55 the authors contrast pyrolysis and hydrochar production (hydrothermal carbonization) but in the abstract, they claim hydrochar was produced by pyrolysis. Clarify.

Line 71, delete and absorption kinetics

Preparation of hydrochar

The collection or source of cornstalk is not mentioned

Batch equilibrium studies

The model of pH measuring device is missing

Line 96, is the initial concentrations denoted by C_e ? Are concentrations listed in bracket equilibrium concentrations?

Line 100, was the adsorption capacity data the only data analyzed using Origin software?

Results and discussion

Line 140, was FTIR performed during the process and at what time periods, or after hydrothermal carbonization?

Is there a reference to support the claim on lines 157-159 on these wavebands?

Line 167, replace "the same" with constant

Line 168-171 the author presents contradictory statements. Maximum adsorption capacity (3.01 at pH 3) and minimum (0.5 at pH) but authors also state that adsorption capacity increased from pH 3 to 9.

Line 178-179, the authors suggest that there is no significant difference between the adsorption capacity at pH 9 and pH 11. Is this supported? What is the measure of the stability referred to? Clarify

If the pKa is 3.2 what is nature (charge) of the mesotrione molecule at $\text{pH} < 3.2$ (at pH 3) and $\text{pH} > 3.2$ (5-11) and does this influence the observed trend?

Line 181-183 the authors make reference to clofencet and salicylic acid pesticides which are not structurally related to the compound under study. Additionally, the study is on adsorption on soil which is very different from hydrochars. Is this citation helpful? Clarify

Line 190- is 0-20 mg/L initial or equilibrium concentrations?

Line 194-196 the authors state that adsorption is a spontaneous exothermic process? Is this statement objectively true? Are there no endothermic adsorption processes? Furthermore, lines 193-194 states the "adsorption of mesotrione was not affected by temperature". Is this what is expected for an exothermic process? The authors then again state that the adsorption capacity decreased with an increase in temperature. These are contradictory statements. The authors should perform a statistical analysis of the variations observed to determine if the values are significantly different or not.

Line 196: How did they arrive at the statement that the process is physisorption and not otherwise?

Lines 199-204: Can the authors find relevant materials such as biochars or carbonaceous materials for comparison?

Figure 4 should be enlarged for improved visibility

Lines 208-209: The reference 27 should be replaced by the original papers for Quasi-first-order and quasi-second-order kinetic models. These equations pre-existed reference 27.

Lines 213-215: Its necessary to indicate whether the experimental conditions were the same for this comparison between hydrochar and biochar for proper interpretation

Line 234: Is the unit for $k_2 \text{ min}^{-1}$?

Delete Lines 242-244 on R^2 values. The information is in the Table

What does fitting of the data to the quasi-second-order kinetic model mean for the adsorption process under study? An interpretation is lacking

Line 268: References for these adsorption isotherms are missing

The isotherms are presented in the text in their nonlinear form yet the data was processed using linear regressions. The equations should be presented in the linear form used.

Figure 6 should be enlarged. The data in the graphs are not legible

The line to 268 mentions linear regression coefficients, the equations presented are nonlinear, the figures plotted are linear and line 288 mentions that nonlinear regression coefficients are reported in the Table. This is confusing. What method was used in data analysis?

Line 290-291- Error functions such as RMSE should be used to determine best fitting isotherm model just as in kinetics

Line 299-300: The specific surfaces areas and CECs mentioned have not been presented anywhere in the manuscript

Line 312 indicates the adsorption is chemical while line 196 indicated the adsorption is physical. Clarify

Is it necessary to discuss the Temkin model if it was not the best fitting?

Line 330: The units for distribution coefficient is lacking

Line 340-341 indicates the adsorption process is endothermic with a positive enthalpy value.

However, in lines 194-196 the authors stated that the adsorption was exothermic. Can this contradiction be clarified? Can this be harmonized with the variation in adsorption capacity with temperature?

Equation 8 used the distribution coefficient which should have units (probably L/g). However, the units of ΔG in Table 3 is kJ/mol. If the parameters in equation 8 (R, T and $\ln K$) are multiplied their units don't cancel out to give kJ/mol. The implication is $\ln K$ must be dimensionless. The authors should convert the distribution coefficient to a dimensionless function and then determine ΔG . This will also affect the values used in equation 9 and the reported values in table 3 are in doubt. Thermodynamics parameters should be recalculated.

Review form: Reviewer 3

Is the manuscript scientifically sound in its present form?

No

Are the interpretations and conclusions justified by the results?

No

Is the language acceptable?

No

Do you have any ethical concerns with this paper?

No

Have you any concerns about statistical analyses in this paper?

No

Recommendation?

Reject

Comments to the Author(s)

The authors reported "Preparation and characterization of cornstalks microspheric hydrochar and adsorption mechanism of mesotrione". After taking consideration, I do not suggest it for publication in this journal. My comments are listed as follows:

1. I have prepared more hydrochar samples from lignocellulose materials. The microsphere is possible, but the microsphere is often not too much (~20%), not total compared to pure carbohydrates such as glucose. The SEM image only confirmed some part of the morphology of hydrochar, not total.
2. The main problem of using hydrochar for water treatment is to contribute to increase the TOC of water. The water after adsorption often has slight dark colour because the release of dissolve organic compounds from hydrochar.
3. The authors missed some important properties of the materials (i.e., zeta potential, BET surface area).
4. Many important adsorption data were missed such as effect of NaCl, desorption study.....
5. Adsorption mechanism needs to be proposed based on the experimental data, not from the literature.
6. The problem of using the liner method for calculating the parameters of kinetics and isotherm models has intensively discussed in the literature. The authors repeated this problem.

7. In the same data, but the authors represented different figures (Figures 4, 6, and 7). This is defined as redundant presentation and should be avoided in any ISI paper.
8. The authors only concluded the data of adsorption were described by models (i.e., PSO) without any further giving information and strong discussion.
9. The authors did not prepare this work carefully. As a result, there are a lot of mistakes, such as KL (L/g) in Table 2? k2 (min⁻¹) page 234 ? Quasi-first-order????????????? Quasi-second-order ????
10. There are a lot of uncertain or incorrect conclusions, such as
 - + "The physical adsorption and desorption rates were extremely fast and were not generally affected by temperature." this is not true for all cases.
 - + "FT-IR spectra showed that the HC contained a large number of -OH groups, C=C bonds of aromatic rings, C-H groups in aromatic rings and phenolic C-O bonds." Any hydrochar material contains those functional groups. The large number of those groups cannot be obtained by FTIR.
 - + "The positive value of ΔH indicated an endothermic process between the adsorbate and the adsorbent [35]. Mesotrione molecules entered the surface or interlayer of the hydrochar from the aqueous phase"?????????????
 - + "The maximum adsorption capacities of the hydrochar at equilibrium were 1.947, 3.015, 3.442, 3.564, and 3.694 mg g⁻¹, corresponding to initial mesotrione concentrations of 10, 20, 30, 40, and 50 mg L⁻¹, respectively.????????????????????????????????? The information is repeated the data from Figure? and they are also not called as the maximum adsorption capacities.

Decision letter (RSOS-202209.R0)

Dear Professor Yang:

Title: Preparation and characterization of cornstalk microspheric hydrochar and adsorption mechanism of mesotrione
 Manuscript ID: RSOS-202209

The editor assigned to your manuscript has now received comments from reviewers. We would like you to revise your paper in accordance with the referee and Subject Editor suggestions which can be found below (not including confidential reports to the Editor). Please note this decision does not guarantee eventual acceptance. I apologise that it has taken longer than usual to be able to send you this decision.

Please submit your revised paper before 13-May-2021. Please note that the revision deadline will expire at 00.00am on this date. If we do not hear from you within this time then it will be assumed that the paper has been withdrawn. In exceptional circumstances, extensions may be possible if agreed with the Editorial Office in advance. We do not allow multiple rounds of revision so we urge you to make every effort to fully address all of the comments at this stage. If deemed necessary by the Editors, your manuscript will be sent back to one or more of the original reviewers for assessment. If the original reviewers are not available we may invite new reviewers.

On behalf of the Subject Editor Professor Anthony Stace and the Associate Editor Dr Dattatray Late.

RSC Associate Editor:
Comments to the Author:
Major Revision

RSC Subject Editor:
Comments to the Author:
(There are no comments.)

Reviewers' Comments to Author:
Reviewer: 1

Comments to the Author(s)

Yang and co-workers reported a mild hydrothermal method to acquire the microspherical structure of hydrochar from cornstalk and investigated the adsorption capacity of hydrochar. The article is well organized and the presentation is also good. However, some minor issues still need to be improved:

1. For the SEM images, the figure 1(c) should provide with the same magnification and scale compared with figure 1(a)(b).
2. The adsorption capacity and removal rate decreased or tended to become stable when $\text{pH} > 9$. Please illustrate why the optimum initial $\text{pH} = 7$ was selected for the subsequent experiments.

Reviewer: 2

Comments to the Author(s)

Grammatical mistakes and spelling:

post-emergence line 46

The claim in line 47 requires reference

The thoughts on line 47 and 48 needs proper connection

The threat of the compound's presence in water has not been highlighted

In line 55 the authors contrast pyrolysis and hydrochar production (hydrothermal carbonization) but in the abstract, they claim hydrochar was produced by pyrolysis. Clarify.

Line 71, delete and absorption kinetics

Preparation of hydrochar

The collection or source of cornstalk is not mentioned

Batch equilibrium studies

The model of pH measuring device is missing

Line 96, is the initial concentrations denoted by C_e ? Are concentrations listed in bracket equilibrium concentrations?

Line 100, was the adsorption capacity data the only data analyzed using Origin software?

Results and discussion

Line 140, was FTIR performed during the process and at what time periods, or after hydrothermal carbonization?

Is there a reference to support the claim on lines 157-159 on these wavebands?

Line 167, replace "the same" with constant

Line 168-171 the author presents contradictory statements. Maximum adsorption capacity (3.01 at pH 3) and minimum (0.5 at pH) but authors also state that adsorption capacity increased from pH 3 to 9.

Line 178-179, the authors suggest that there is no significant difference between the adsorption capacity at pH 9 and pH 11. Is this supported? What is the measure of the stability referred to? Clarify

If the pK_a is 3.2 what is nature (charge) of the mesotrione molecule at $pH < 3.2$ (at pH 3) and $pH > 3.2$ (5-11) and does this influence the observed trend?

Line 181-183 the authors make reference to clofencet and salicylic acid pesticides which are not structurally related to the compound under study. Additionally, the study is on adsorption on soil which is very different from hydrochars. Is this citation helpful? Clarify

Line 190- is 0-20 mg/L initial or equilibrium concentrations?

Line 194-196 the authors state that adsorption is a spontaneous exothermic process? Is this statement objectively true? Are there no endothermic adsorption processes? Furthermore, lines 193-194 states the "adsorption of mesotrione was not affected by temperature". Is this what is expected for an exothermic process? The authors then again state that the adsorption capacity decreased with an increase in temperature. These are contradictory statements. The authors should perform a statistical analysis of the variations observed to determine if the values are significantly different or not.

Line 196: How did they arrive at the statement that the process is physisorption and not otherwise?

Lines 199-204: Can the authors find relevant materials such as biochars or carbonaceous materials for comparison?

Figure 4 should be enlarged for improved visibility

Lines 208-209: The reference 27 should be replaced by the original papers for Quasi-first-order and quasi-second-order kinetic models. These equations pre-existed reference 27.

Lines 213-215: Its necessary to indicate whether the experimental conditions were the same for this comparison between hydrochar and biochar for proper interpretation

Line 234: Is the unit for k_2 min^{-1} ?

Delete Lines 242-244 on R^2 values. The information is in the Table

What does fitting of the data to the quasi-second-order kinetic model mean for the adsorption process under study? An interpretation is lacking

Line 268: References for these adsorption isotherms are missing

The isotherms are presented in the text in their nonlinear form yet the data was processed using linear regressions. The equations should be presented in the linear form used.

Figure 6 should be enlarged. The data in the graphs are not legible

The line to 268 mentions linear regression coefficients, the equations presented are nonlinear, the figures plotted are linear and line 288 mentions that nonlinear regression coefficients are reported in the Table. This is confusing. What method was used in data analysis?

Line 290-291- Error functions such as RMSE should be used to determine best fitting isotherm model just as in kinetics

Line 299-300: The specific surfaces areas and CECs mentioned have not been presented anywhere in the manuscript

Line 312 indicates the adsorption is chemical while line 196 indicated the adsorption is physical. Clarify

Is it necessary to discuss the Temkin model if it was not the best fitting?

Line 330: The units for distribution coefficient is lacking

Line 340-341 indicates the adsorption process is endothermic with a positive enthalpy value.

However, in lines 194-196 the authors stated that the adsorption was exothermic. Can this contradiction be clarified? Can this be harmonized with the variation in adsorption capacity with temperature?

Equation 8 used the distribution coefficient which should have units (probably L/g). However, the units of ΔG in Table 3 is kJ/mol . If the parameters in equation 8 (R , T and $\text{Ln}K$) are multiplied their units don't cancel out to give kJ/mol . The implication is $\text{Ln}K$ must be dimensionless. The authors should convert the distribution coefficient to a dimensionless function and then determine ΔG . This will also affect the values used in equation 9 and the reported values in table 3 are in doubt. Thermodynamics parameters should be recalculated.

Reviewer: 3

Comments to the Author(s)

The authors reported "Preparation and characterization of cornstalks microspheric hydrochar and adsorption mechanism of mesotrione". After taking consideration, I do not suggest it for publication in this journal. My comments are listed as follows:

1. I have prepared more hydrochar samples from lignocellulose materials. The microsphere is possible, but the microsphere is often not too much (~20%), not total compared to pure carbohydrates such as glucose. The SEM image only confirmed some part of the morphology of hydrochar, not total.
2. The main problem of using hydrochar for water treatment is to contribute to increase the TOC of water. The water after adsorption often has slight dark colour because the release of dissolve organic compounds from hydrochar.
3. The authors missed some important properties of the materials (i.e., zeta potential, BET surface area).
4. Many important adsorption data were missed such as effect of NaCl, desorption study.....
5. Adsorption mechanism needs to be proposed based on the experimental data, not from the literature.
6. The problem of using the liner method for calculating the parameters of kinetics and isotherm models has intensively discussed in the literature. The authors repeated this problem.
7. In the same data, but the authors represented different figures (Figures 4, 6, and 7). This is defined as redundant presentation and should be avoided in any ISI paper.

8. The authors only concluded the data of adsorption were described by models (i.e., PSO) without any further giving information and strong discussion.
9. The authors did not prepare this work carefully. As a result, there are a lot of mistakes, such as KL (L/g) in Table 2? k_2 (min⁻¹) page 234 ? Quasi-first-order?? Quasi-second-order ??
10. There are a lot of uncertain or incorrect conclusions, such as
 + “The physical adsorption and desorption rates were extremely fast and were not generally affected by temperature.” this is not true for all cases.
 + “FT-IR spectra showed that the HC contained a large number of -OH groups, C=C bonds of aromatic rings, C-H groups in aromatic rings and phenolic C-O bonds.” Any hydrochar material contains those functional groups. The large number of those groups cannot be obtained by FTIR.
 + “The positive value of ΔH indicated an endothermic process between the adsorbate and the adsorbent [35]. Mesotrione molecules entered the surface or interlayer of the hydrochar from the aqueous phase”??
 + “The maximum adsorption capacities of the hydrochar at equilibrium were 1.947, 3.015, 3.442, 3.564, and 3.694 mg g⁻¹, corresponding to initial mesotrione concentrations of 10, 20, 30, 40, and 50 mg L⁻¹, respectively.?? The information is repeated the data from Figure? and they are also not called as the maximum adsorption capacities.

Author's Response to Decision Letter for (RSOS-202209.R0)

See Appendix A.

Decision letter (RSOS-202209.R1)

Dear Professor Yang:

Title: Preparation and characterization of cornstalk microspheric hydrochar and adsorption mechanism of mesotrione
 Manuscript ID: RSOS-202209.R1

Thank you for submitting the above manuscript to Royal Society Open Science. On behalf of the Editors and the Royal Society of Chemistry, I am pleased to inform you that your manuscript will be accepted for publication in Royal Society Open Science subject to minor revision in accordance with the referee suggestions. Please find the reviewers' comments at the end of this email.

The reviewers and handling editors have recommended publication, but also suggest some minor revisions to your manuscript. Therefore, I invite you to respond to the comments and revise your manuscript.

Because the schedule for publication is very tight, it is a condition of publication that you submit the revised version of your manuscript before 21-May-2021. Please note that the revision deadline will expire at 00.00am on this date. If you do not think you will be able to meet this date please let me know immediately.

Kind regards,
Dr Laura Smith
Publishing Editor, Journals

On behalf of the Subject Editor Professor Anthony Stace and the Associate Editor Dr Dattatray Late.

RSC Associate Editor

Comments to the Author:

Remove the background colour in the figure 2 Fourier transform infrared spectroscopy (FTIR) characterization of hydrochar (HC).

Reviewer comments to Author:

Author's Response to Decision Letter for (RSOS-202209.R1)

See Appendix B.

Decision letter (RSOS-202209.R2)

Dear Professor Yang:

Title: Preparation and characterization of cornstalk microspheric hydrochar and adsorption mechanism of mesotrione

Manuscript ID: RSOS-202209.R2

It is a pleasure to accept your manuscript in its current form for publication in Royal Society Open Science. The chemistry content of Royal Society Open Science is published in collaboration with the Royal Society of Chemistry.

Yours sincerely,

Dr Laura Smith

Publishing Editor, Journals

On behalf of the Subject Editor Professor Anthony Stace and the Associate Editor Dr Dattatray Late.

RSC Associate Editor
Comments to the Author:
Accept as is

Reviewer(s)' Comments to Author:

Appendix A

Response to Reviewers

Dear Reviewer 1:

Thank you for taking so much time on reviewing my manuscript and giving me so many sincere comments. I have carefully revised all your suggestions as follows. Substantia revision were made (**Please reference to "Red marked in the revised manuscript"**).

The problem (1) : For the SEM images, the figure 1(c) should provide with the same magnification and scale compared with figure 1(a)(b).

Answer: Thanks for your good advice, the same magnification and scale figure were provided in the revised manuscript.

The problem (2) : The adsorption capacity and removal rate decreased or tended to become stable when $\text{pH} > 9$. Please illustrate why the optimum initial $\text{pH} = 7$ was selected for the subsequent experiments.

Answer: The adsorption capacity and removal rate rapidly increased when the initial solution pH rose from 3 to 9. So, the pH value can influence the adsorption capacity and removal rate significantly. Finally, the neutral pH value with relatively mild experimental conditions was selected as the condition for subsequent experiments.

Dear Reviewer 2:

Thank you for taking time out of your busy schedule to review my manuscript and give me the sincere comment. I have carefully revised all your suggestions as

follows. Substantia revision were made (**Please reference to "Blue marked in the revised manuscript"**).

The problem (1) : post-emergence line 46.

Answer: Thanks for your good advice, the spelling mistake was carefully modified in the revised manuscript.

The problem (2) : The claim in line 47 requires reference .

Answer: Thanks for your good advice, the reference was added in the revised manuscript. (Rouchaud, J., Neus, O., Eelen, H., & Bulcke, R. (2000). Dissipation and mobility of the herbicide mesotrione in the soil of corn crops. mededelingen.)

The problem (3) : The thoughts on line 47 and 48 needs proper connection.

Answer: Thanks for your good advice, the thoughts on line 47 and 48 was carefully modified in the revised manuscript.

The problem (4) : The threat of the compound's presence in water has not been highlighted.

Answer: Thanks for your good advice, the threat of the compound's presence in water were added in the revised manuscript (line 50-52).

The problem (5) : In line 55 the authors contrast pyrolysis and hydrochar production (hydrothermal carbonization) but in the abstract, they claim hydrochar was produced by pyrolysis. Clarify.

Answer: It has been carefully revised (line 30-31).

The problem (6) : Line 71, delete and absorption kinetics.

Answer: Thanks for your good advice, "and absorption kinetics" were deleted

in the revised manuscript.

The problem (7) : The collection or source of cornstalk is not mentioned.

Answer: Thanks for your good advice, “The collection or source of cornstalk” were added in the revised manuscript (line 75-76). “The cornstalks were collected from Agricultural Science Experimental Station of Jilin Agricultural University.”

The problem (8) : The model of pH measuring device is missing.

Answer: Thanks for your good advice, “model of pH measuring device” was added in the revised manuscript (line 95).

The problem (9) : Line 96, is the initial concentrations denoted by C_e ? Are concentrations listed in bracket equilibrium concentrations?

Answer: Thanks, the mistakes were correct in the revised manuscript (line 98-99).

The problem (10) : Line 100, was the adsorption capacity data the only data analyzed using Origin software?

Answer: Thanks, the mistakes were correct in the revised manuscript (line 102-103).

The problem(11): Line 140, was FTIR performed during the process and at what time periods, or after hydrothermal carbonization?

Answer: Thanks, the mistakes were correct in the revised manuscript (line 141-142).

The problem (12) : Is there a reference to support the claim on lines 157-159 on these wavebands?

Answer: Thanks for your good advice, the reference was added in the revised manuscript. (Deng Congjing, Ma Huanhuan, Wang Liangcai, Zhu Zhengxiang, Zhou Jianbin. 2019 Structure Characterization and Pyrolysis Properties of Apricot Shell Hemicellulose. SCIENTIA SILVAE SINICAE,55(1): 74-80.)

The problem (13) : Line 167, replace “the same” with constant.

Answer: Thanks, the mistakes were correct in the revised manuscript (line 169).

The problem (14) : Line 168-171 the author presents contradictory statements. Maximum adsorption capacity (3.01 at pH 3) and minimum (0.5 at pH) but authors also state that adsorption capacity increased from pH 3 to 9.

Answer: Thanks, the mistakes were correct in the revised manuscript (line 171-173).

The problem (15) : Line 178-179, the authors suggest that there is no significant difference between the adsorption capacity at pH 9 and pH 11. Is this supported? What is the measure of the stability referred to? Clarify.

Answer: Thanks, This statement is of no substance and has been deleted in the revised manuscript.

The problem (16) : If the pKa is 3.2 what is nature (charge) of the mesotrione molecule at pH<3.2 (at pH 3) and pH>3.2 (5-11) and does this influence the observed trend?

Answer: Thanks, This statement is of no substance and has been deleted in the revised manuscript.

The problem (17) : Line 181-183 the authors make reference to clofencet and salicylic acid pesticides which are not structurally related to the compound under study. Additionally, the study is on adsorption on soil which is very different from hydrochars. Is this citation helpful? Clarify

Answer: Thanks, This statement is of no substance and has been deleted in the revised manuscript.

The problem (18) : Line 190- is 0-20 mg/L initial or equilibrium concentrations?

Answer: Thanks, 0-20 mg/L is initial concentrations, and has been carefully modified in the revised manuscript.

The problem (19) : Line 194-196 the authors state that adsorption is a spontaneous exothermic process? Is this statement objectively true? Are there no endothermic adsorption processes? Furthermore, lines 193-194 states the “adsorption of mesotrione was not affected by temperature”. Is this what is expected for an exothermic process? The authors then again state that the adsorption capacity decreased with an increase in temperature. These are contradictory statements. The authors should perform a statistical analysis of the variations observed to determine if the values are significantly different or not.

Answer: Thanks for your good advice, the results are discussed again in the revised manuscript. (line 184-192).

The problem (20) : Line 196: How did they arrive at the statement that the process is physisorption and not otherwise?

Answer: In this part, only the influence of temperature on the adsorption

of mesotrione is discussed, and the physical adsorption can not be obtained, and has been deleted in the revised manuscript.

The problem (21) : Lines 199-204: Can the authors find relevant materials such as biochars or carbonaceous materials for comparison?

Answer: Thanks for your good advice, the reference was added in the revised manuscript. (**Xiao Yang. Ethylenethiourea and mesotrione in wastewater: the determination using HPLC and removal by biochar [D]. Shandong: Ocean University of China,2014.**)

The problem (22) : Figure 4 should be enlarged for improved visibility.

Answer: Thanks, the Figure 4 was enlarged in the revised manuscript.

The problem (23) : Lines 208-209: The reference 27 should be replaced by the original papers for Quasi-first-order and quasi-second-order kinetic models. These equations pre-existed reference 27.

Answer: There is no need to cite references, as Quasi-first-order and quasi-second-order kinetic models were shown detailly in the article (line 212-223).

The problem (24) : Lines 213-215: Its necessary to indicate whether the experimental conditions were the same for this comparison between hydrochar and biochar for proper interpretation.

Answer: Thanks, This statement is of no substance and has been deleted in the revised manuscript.

The problem (25) : Line 234: Is the unit for $k_2 \text{ min}^{-1}$?

Answer: Thanks, the mistakes were correct in the revised manuscript (line

231-232).

The problem (26) : Delete Lines 242-244 on R^2 values. The information is in the Table.

Answer: Thanks, Lines 242-244 on R^2 values were deleted in the revised manuscript.

The problem (27) : What does fitting of the data to the quasi-second-order kinetic model mean for the adsorption process under study? An interpretation is lacking.

Answer: Among the kinetic models used, the quasi-second-order model was the best with the largest R^2 (0.999), and the equilibrium adsorption value calculated by the model was extremely close to the experimental values. Therefore, the quasi-second-order model best described the adsorption process of mesotrione onto hydrochar.

The problem (28) : Line 268: References for these adsorption isotherms are missing .

Answer: There is no need to cite references, as Langmuir, Freundlich and Temkin models were shown detaily in the article (line 255-266).

The problem (29) : The isotherms are presented in the text in their nonlinear form yet the data was processed using linear regressions. The equations should be presented in the linear form used.

Answer: Thanks, It was carefully revised in the revised manuscript.

The problem (30) : Figure 6 should be enlarged. The data in the graphs are not legible

Answer: Thanks, the Figure 6 was enlarged in the revised manuscript.

The problem (31) : The line to 268 mentions linear regression coefficients, the equations presented are nonlinear, the figures plotted are linear and line 288 mentions that nonlinear regression coefficients are reported in the Table. This is confusing. What method was used in data analysis?

Answer: Thanks, It was carefully revised in the revised manuscript. The data analysis method can be reflected in the picture 6 of the article.

The problem(32): Line 290-291- Error functions such as RMSE should be used to determine best fitting isotherm model just as in kinetics.

Answer: Thanks, It was carefully revised in the revised manuscript.

The problem(33): Line 299-300: The specific surfaces areas and CECs mentioned have not been presented anywhere in the manuscript .

Answer: Thanks, the specific surfaces areas and CECs mentioned were deleted in the revised manuscript.

The problem (34) :Line 312 indicates the adsorption is chemical while line 196 indicated the adsorption is physical. Clarify, Is it necessary to discuss the Temkin model if it was not the best fitting?

Answer: The whole adsorption process was based on physical adsorption, with chemical adsorption as a supplement. It is necessary to discuss the Temkin model although it was not the best fitting.

The problem (35) : Line 330: The units for distribution coefficient is lacking .

Answer: Thanks, The units for distribution coefficient was added.

The problem (36) : Line 340-341 indicates the adsorption process is endothermic with a positive enthalpy value. However, in lines 194-196 the authors stated that the adsorption was exothermic. Can this contradiction be clarified? Can this be harmonized with the variation in adsorption capacity with temperature? .

Answer: It was carefully revised and can be harmonized with the variation in adsorption capacity with temperature in the revised manuscript.

The problem (34) : Equation 8 used the distribution coefficient which should have units (probably L/g). However, the units of ΔG in Table 3 is kJ/mol. If the parameters in equation 8 (R , T and $\ln K$) are multiplied their units don't cancel out to give kJ/mol. The implication is $\ln K$ must be dimensionless. The authors should convert the distribution coefficient to a dimensionless function and then determine ΔG . This will also affect the values used in equation 9 and the reported values in table 3 are in doubt. Thermodynamics parameters should be recalculated. .

Answer: Thanks, It was carefully revised in the revised manuscript.

Dear Reviewer 3:

Thank you for taking time out of your busy schedule to review my manuscript and give me the sincere comment. I have carefully revised all your suggestions as follows. Substantia revision were made (**Please reference to "green marked in the revised manuscript"**).

The problem (1): I have prepared more hydrochar samples from lignocellulose materials. The microsphere is possible, but the microsphere is often not too much

(~20%), not total compared to pure carbohydrates such as glucose. The SEM image only confirmed some part of the morphology of hydrochar, not total.

Answer: Thanks, The morphology and quantity of hydrothermal carbon microspheres prepared at different temperatures were different. The surface microspheres of corn stalk hydrothermal carbon prepared at 240 °C were relatively rich in structure though SEM image only confirmed some part of the morphology of hydrochar.

The problem (2): The main problem of using hydrochar for water treatment is to contribute to increase the TOC of water. The water after adsorption often has slight dark colour because the release of dissolve organic compounds from hydrochar.

Answer: Thank you for your detailed interpretation.

The problem (3): The authors missed some important properties of the materials (i.e., zeta potential, BET surface area).

Answer: Thank you for your good advices. Influenced by the COVID-19 epidemic, the measurement of many indicators has been limited by conditions.

The problem (4): Many important adsorption data were missed such as effect of NaCl, desorption study.....

Answer: These aspects will be constantly improved in the following research.

The problem (5): Adsorption mechanism needs to be proposed based on the experimental data, not from the literature.

Answer: Thanks, It was carefully revised in the revised manuscript.

The problem (6): The problem of using the liner method for calculating the parameters of kinetics and isotherm models has intensively discussed in the literature. The authors repeated this problem.

Answer: Thanks, It was carefully revised in the revised manuscript.

The problem (7): In the same data, but the authors represented different figures (Figures 4, 6, and 7). This is defined as redundant presentation and should be avoided in any ISI paper.

Answer: Thanks, It was carefully revised in the revised manuscript.

The problem (8): The authors only concluded the data of adsorption were described by models (i.e., PSO) without any further giving information and strong discussion.

Answer: Thanks, It was carefully revised in the revised manuscript.

The problem (9): The authors did not prepare this work carefully. As a result, there are a lot of mistakes, such as KL (L/g) in Table 2? k2 (min⁻¹) page 234 ? Quasi-first-order?? Quasi-second-order ??

Answer: Thanks, It was carefully revised in the revised manuscript.

The problem (10): There are a lot of uncertain or incorrect conclusions, such as

+ “The physical adsorption and desorption rates were extremely fast and were not generally affected by temperature.” this is not true for all cases.

+ “FT-IR spectra showed that the HC contained a large number of -OH groups, C=C bonds of aromatic rings, C-H groups in aromatic rings and

phenolic C-O bonds.” Any hydrochar material contains those functional groups. The large number of those groups cannot be obtained by FTIR.

+ “The positive value of ΔH indicated an endothermic process between the adsorbate and the adsorbent [35]. Mesotrione molecules entered the surface or interlayer of the hydrochar from the aqueous phase”??

+ “The maximum adsorption capacities of the hydrochar at equilibrium were 1.947, 3.015, 3.442, 3.564, and 3.694 mg g⁻¹, corresponding to initial mesotrione concentrations of 10, 20, 30, 40, and 50 mg L⁻¹, respectively.?? The information is repeated the data from Figure? and they are also not called as the maximum adsorption capacities.

Answer: Thanks, It was carefully revised in the revised manuscript. Especially after revising the suggestions given by reviewer 2, many existing problems have been solved.

Appendix B

Response

Dear Editor:

Thank you for taking so much time on reviewing my manuscript and giving me so many sincere comments. I have carefully revised your suggestions as follows.

The problem: Remove the background colour in the figure 2 Fourier transform infrared spectroscopy (FTIR) characterization of hydrochar (HC)..

Answer: Thanks for your good advice, the background colour in the figure 2 were removed in the revised manuscript.